# PIDS: A User-Friendly Plant DNA Fingerprint Database Management System

**DOI:** 10.3390/genes11040373

**Published:** 2020-03-30

**Authors:** Bin Jiang, Yikun Zhao, Hongmei Yi, Yongxue Huo, Haotian Wu, Jie Ren, Jianrong Ge, Jiuran Zhao, Fengge Wang

**Affiliations:** Maize Research Center, Beijing Academy of Agriculture and Forestry Sciences/Beijing Key Laboratory of Maize DNA Fingerprinting and Molecular Breeding, Beijing 100097, China; j13311151986@gmail.com (B.J.); zhaoqiankaisteam@126.com (Y.Z.); yhmmaize@163.com (H.Y.); holibut@163.com (Y.H.); nghsky@163.com (H.W.); renjie@maizedna.org (J.R.); 13439992517@163.com (J.G.); maizezhao@126.com (J.Z.)

**Keywords:** microsatellites, database, DNA fingerprint, algorithms, genotyping

## Abstract

The high variability and somatic stability of DNA fingerprints can be used to identify individuals, which is of great value in plant breeding. DNA fingerprint databases are essential and important tools for plant molecular research because they provide powerful technical and information support for crop breeding, variety quality control, variety right protection, and molecular marker-assisted breeding. Building a DNA fingerprint database involves the production of large amounts of heterogeneous data for which storage, analysis, and retrieval are time and resource consuming. To process the large amounts of data generated by laboratories and conduct quality control, a database management system is urgently needed to track samples and analyze data. We developed the plant international DNA-fingerprinting system (PIDS) using an open source web server and free software that has automatic collection, storage, and efficient management functions based on merging and comparison algorithms to handle massive microsatellite DNA fingerprint data. PIDS also can perform genetic analyses. This system can match a corresponding capillary electrophoresis image on each primer locus as fingerprint data to upload to the server. PIDS provides free customization and extension of back-end functions to meet the requirements of different laboratories. This system can be a significant tool for plant breeders and can be applied in forensic science for human fingerprint identification, as well as in virus and microorganism research.

## 1. Introduction

DNA fingerprints have multi-loci, high variability, and simple and stable inheritance, and have attracted much attention because of their great practical value [1,2]. The high variability and somatic stability of DNA fingerprints can be used to identify individuals, which is of great value in determining kinship among individuals and in forensics to identify criminals [3,4,5,6]. In other applications, DNA fingerprints can provide accurate identification at the varieties level [7,8,9], and play important roles in plant breeding [10,11,12]. DNA fingerprints are also commonly used in research on animal evolutionary trends and animal varieties identification [8,13].

Simple sequence repeats (SSRs) are molecular markers that have the advantages of simplicity, rapidity, high repeatability, and abundant polymorphisms [14,15,16]. SSRs have been used successfully to construct DNA fingerprints that are used widely in plant genetic studies and crop breeding [9,11]. The identification of varieties relies heavily on accurate plant genotype indicators to ensure the genetic and physiological consistency of improved crop varieties [17]. Compared with traditional field planting identification, the analysis of SSR fingerprints avoids environmental interference and can quickly and accurately identify varieties without the restrictions of growth and development times [18,19,20].

DNA fingerprints can be managed systematically by a computer, and can be organized in DNA fingerprint databases [21]. DNA fingerprint databases are essential and important tools for plant molecular research because they provide powerful technical and information support for crop breeding, variety quality control, variety right protection, and molecular marker-assisted breeding [22,23]. Building a DNA fingerprint database involves the production of large amounts of heterogeneous data for which storage, analysis, and retrieval are time and resource consuming [24,25,26,27]. Some biological data management software has been developed. For example, SLIMS can organize, store, and access sample information [28]; AutoLabDB provides database schema to support automated laboratories [29]; and TheSNPpit can manage large amounts of multi-panel SNP genotype data [30].

However, no automatic management system for plant SSR fingerprints from “experiment to storage to fingerprint audit to analysis” is available, and no database management system for plant SSR fingerprint development has been reported so far. The construction of an SSR fingerprint database for plant varieties is particularly difficult for a number of reasons. First, plant variety is a concept that can have different meanings. For example, plant DNA fingerprint databases can have two sampling methods, mixed and individual samples, which is unlike a human DNA fingerprint database, which will have only individual samples [1,3,4,9]. Therefore, a management system of plant DNA fingerprint data must be able to merge the fingerprints of multiple individual samples, individual and mixed samples, and mixed samples [9,10,30]. Second, plant varieties tend to have poor consistency between individual samples [9,10]. When collecting DNA fingerprints of mixed samples, it is difficult to describe high and low peaks, and three or multiple peaks quantitatively because of the low consistency of samples. Therefore, the image of a DNA fingerprint is very important, so after collecting the fingerprint data, it is necessary to upload and maintain a large database of fingerprint images (gel electrophoresis or capillary electrophoresis (CE) image on each primer locus). Third, to select primers for the SSR fingerprint database, high polymorphism was considered as an important selection index, so a large number of primers of the 2-bp repeat type were selected. Because the difference between alleles is small, a naming scheme based on the number of repeats cannot be used when collecting the original data. Therefore, alleles need to be named according to segment size, which can lead to data reading errors of 1–2 bp. To reduce the influence of data acquisition error, a base offset parameter needs to be introduced into data merge and comparison algorithms. Finally, large numbers of SSR primers are commonly used to establish crop variety databases, and a composite amplification system has not yet been developed. Therefore, it is not possible to build a DNA fingerprint database for plants with only a group of panels, as has been done for humans. Instead, multiple panels are needed to complete a plant DNA fingerprint database. This raises the problem of how to combine data in multiple panels into data similar to a group of panels to facilitate the management and query of test information. Finally, the current biological data management systems do not solve the fingerprint processing problem of data from repeated experiments. Therefore, the development of a suitable plant variety SSR fingerprint database management system is very necessary.

In this paper, we describe the Plant International DNA-Fingerprinting System (PIDS) that we developed to solve the problems created by the situations described in the previous paragraph. PIDS has automatic collection, storage, and efficient management functions based on merging and comparison algorithms to handle massive amounts of fingerprint data, and the system can also perform genetic analyses. PIDS is available online at https://ssr.PIDS.online:8445/ssr. This system can match a corresponding CE image on each primer locus as fingerprint data to upload to the server. For more than five years, PIDS has been used widely by 20 institutions, including the China National Rice Research Institute, the Chinese Academy of Agricultural Sciences, and the Development Center for Science and Technology (China), and has proved to be a stable efficient system for sharing data and the results of genetic analyses.

## 2. Materials and Methods 

### 2.1. System and Database Schema

In genotyping experiments of individual and population samples, representative fingerprint data should be screened out from the fingerprints of multiple parallel experiments performed by different experimenters and/or from multiple repeated experiments performed by a single experimenter. Therefore, PIDS constructs a hierarchical weighted tree structure (experiment–DNA–experimenter–sample) that is used to quantify and reflect the impact of different experiment stages on the quality of the automated integrated DNA fingerprint data, so as to accurately evaluate and select the best DNA fingerprint, while avoiding manual data errors. This method uses an automated optimal typing and selection algorithm based on DNA fingerprint literal data. For any problematic data, manual verification and genotype reselection is used to generate DNA fingerprint data that meet experimental requirements. In this way, unreliable genotype data that could interfere with the final DNA fingerprint are avoided, so as to improve the efficiency of the whole data integration processing and reduce the occurrence of errors.

The PIDS schema contains hierarchical permission structures that limit the query and operation rights of fingerprint data as follows. An experimenter can query and operate their own uploaded fingerprint data, but cannot access data from others; and after an experimenter uploads a Sample Information Table (SIT) to clarify which samples he/she is responsible for, then the experimenter is able to query and audit the sample fingerprints (in the SIT) from multiple parallel experiments submitted by different experimenters.

The fingerprint data are stored in different fingerprint databases according to their different purposes and functions as follows. Experimental Fingerprint Database (EFD): An experimenter can upload an Excel file, GeneMapper output file, and project file into the EFD. Original fingerprint information is recorded and can be queried and traced. Every piece of fingerprint data in the EFD must be audited through the Fingerprint Merging Algorithm by the experimenter before the fingerprint data are submit automatically to the Sample Fingerprint Database (SFD). This merging algorithm can solve the problem of fingerprint duplication in multiple experiments of a single experimenter and reduce experimental errors. This design also ensures the integrity of data and avoids the absence of loci data. Sample Fingerprint Database (SFD): An experimenter can audit the sample fingerprint data (in the SIT) from the EFD. After the data are audited and confirmed by the experimenter, a set of sample fingerprints are generated and submitted automatically to the Local Fingerprint Database (LFD) using the Fingerprint Merging Algorithm. By merging the audited data, any artificial errors caused by different experimenters can be reduced. The two layers of data audit and merge (EFD–SFD and SFD–LFD) achieve sufficient quality assurance of the experimental results data. Local Fingerprint Database (LFD): The LFD can be used for fingerprint data comparisons and reports. A locking function is provided and, once locked, the data cannot be changed.

PIDS can track DNA samples through workflows, which allows users to trace back to GE and CE files (CE image on each primer locus). Users can also query the sample sources.

### 2.2. Fingerprint Merging and Comparison Algorithms

#### 2.2.1. Fingerprint Merging Algorithm

The procedure used in the Fingerprint Merging Algorithm is shown in Figure 1. The data merging direction is “experiment → DNA → experimenter → sample → variety”, with the aim of achieving optimal selection and generating the final fingerprint data step by step. Four layers and four parameters are defined (Figure 1). The four layers are experiment, DNA, experimenter, and sample. The four parameters are defined as: i, which indicates the number of experiments performed for the current DNA sample by the same experimenter; j, which indicates the number of copies of DNA extracted by the current experimenter from one sample; k, which indicates the number of experimenters assigned to current samples; and m, which indicates the number of samples for the current variety. To treat each sample as an independent variety, m is set as 1.

The algorithm distinguishes the weight of each group of fingerprint data by merging fingerprints step by step, reflects the stage of each fingerprint, and traces the source of fingerprint data errors through each layer by setting weights as follows. The sample layer reflects the differences among different batches of samples and the weight is set as 1. The experimenter layer reflects the data analysis of several experimenters and the weight is set as 1/n. The DNA layer reflects the experimental operation of an experimenter and the weight is set as 1/m. The experiment layer reflects the state of the instrument and the sampling and the weight is set as 1/k. Therefore, the weight accumulation of hierarchical merging preliminarily reflects the qualities of the experiment, instrument, and data analysis of the sample fingerprint. The default base offset during merging is:(1)|x|<2bp

The Fingerprint Merging Algorithm has two categories: peer fingerprint merging and cross-layer fingerprint merging. Peer fingerprint merging refers to the locus-by-locus comparison of multiple sets of data in the same layer. The genotype is retained into the merged result if it is the same among the different datasets, otherwise only the genotype that occurs most frequently is retained.

Cross-layer fingerprint merging is performed as follows:

1. Assign the fingerprint data of the i round of the same DNA solution Ai with the same weight of 1/(i × j × k × m), then perform peer layer fingerprint merging at the experiment layer to obtain the merged fingerprint B of the experiment layer.

2. Assign the merged fingerprint Bj from different DNA solutions of the same sample performed by the same experimenter the same weight of 1/(j × k × m), then perform peer fingerprint merging at the DNA layer to obtain the merged fingerprint C of the DNA layer.

3. Assign the merged fingerprint at the DNA layer Ck of the same sample conducted by different experimenters with the same weight of 1/(k × m), then perform peer layer fingerprint merging at the experimenter layer to obtain the merged fingerprint D of the experimenter layer.

4. Assign the merged fingerprint at the experimenter layer Dm of different samples of the same variety with the same weight of 1/m, then perform peer layer fingerprint merging at the sample layer to obtain the final merged fingerprint of the sample layer, which represents the standard fingerprint of a variety.

5. The cross-layer fingerprint merging is finished.

A variety can be represented by multiple samples, and generally a standard DNA fingerprint database can be constructed using only one of the most authoritative samples to represent one variety. For example, in the DNA fingerprint database of maize, the sample layer reflects the differences among different batches of samples, the experimenter layer reflects the differences of the data analysis layers of several experimenters, the DNA layer reflects the experimental operation layer of the experimenter, and the experiment layer reflects the state of the instruments and sampling methods used. Therefore, through the weight accumulation of hierarchical merging, the quality of the sample fingerprint experiments, instruments, and data analysis can be preliminarily reflected. To reflect the weight of each layer objectively and truthfully, at the laboratory management layer it should be stipulated that different experimenters should not cross-use the same DNA sample and different DNA samples extracted by the same experimenter must be labeled differently, otherwise the system will process them as different repeats of the same DNA. Multiple DNA samples extracted by the same experimenter from a single plant are considered as one repeat regardless of the different DNA numbers.

#### 2.2.2. Fingerprint Comparison Algorithm

Suppose, for a particular crop, two sets of DNA fingerprint data are ready to be compared. They are the pending fingerprint data in queue f(n) and the comparison fingerprint data in queue f(m), where n(n > 0) and m(m > 0) indicate how many DNA fingerprints are in each queue, and each fingerprint is set to contain p(p > 0) loci data. If the number of different loci between the fingerprints in the two queues is D, the number of non-different loci is S, and the number of missing loci is M, then the relationship among them is: (2)p=D+S+M

This equation represents the basic information obtained after a fingerprint comparison. The difference between a fingerprint pair (pending and comparison) can be obtained by defining the proportions of the different loci as:(3)$$x=\frac{D}{p},x\in \left(0,1\right)$$

This equation indicates that the larger the x value is, the greater is the genotyping difference between a fingerprint pair. This value plays an extremely important role in determining the report and other conclusions.

The basic definitions of different loci, non-different loci, and missing loci points are as follows:

1. Number of different loci (D): The total number of differences between the data of two valid loci at the same locus of a fingerprint pair. A threshold value is usually set for the number of different loci to limit the number of comparisons within the expected difference range.

2. Number of missing loci (M): The number of effective loci at the same locus of a fingerprint pair is less than the total number of 2.

3. Number of non-different loci (S): The total number of non-different loci between two valid loci at the same locus of a fingerprint pair. This number can be derived from Equation (2).

Different comparison methods are needed to adapt to different needs. In the Fingerprint Comparison Algorithm, different conditions are used to adjust the comparison methods and the default limit range of the fingerprint data (the fingerprint data that can be used when the fingerprint data range is not specified) needs to be set. If the range of fingerprint data is not specified, the following equation is applied: homonymy + non-homonymy = the entire database. The conditions that need to be specified are listed in Table 1.

The functions of the parameters required in the Fingerprint Comparison Algorithm are described in Table 2.

The pseudocodes for the algorithms are:
Algorithm 1: GCAInput: [a1,b1] as loci 1, [a2,b2] as loci 2, a1/a2 is the first data of diploid, b1/b2 is the lastdata of diploid, the different offset allowed is named nresult = 1 # initialized as not sameconditionR1 = abs(a1 - a2) ≤ nconditionR2 = abs(b1 - b2) ≤ nconditionR3 = abs(a1 - b2) ≤ nconditionR4 = abs(a2 - b1) ≤ nif (conditionR1 and conditionR2) or (conditionR3 and conditionR4)result = 0end ifreturn resultOutput: 0-> two loci are same; 1-> two loci are differentAlgorithm 2: FCPPInput: queue f1 of length n, queue f2 of length m, f1 and f2 should have same number ofloci, named as pInitialize compareResult array as size m*nfor i = 1 to m  for j = 1 to n    diffCount = 0      for k = 1 to p        diffCount=diffCount+GCA(f1[j][p]+f2[i][p])      end for      compareResult.append([f1[j], f2[i], diffCount])   end forend forreturn compareResultOutput: compareResult

### 2.3. System and Database Implementation

PIDS was developed over two years of continuous use and has evolved in parallel with the automation of biological experiments. PIDS is implemented on a Tomcat server (The Apache Software Foundation, Wakefield, MA, USA). The system uses browser/server (B/S) architecture combined with Java and J2EE technologies. The MySQL web server (Oracle Corporation, Redwood Shores, CA, USA) is used for database storage. PIDS is flexible and can be expanded to accommodate multiple detection techniques (e.g., SNPs, indels) and for storage and analysis of multiple fingerprint data. This system is compatible with fingerprint data for DNA testing of all plant varieties. It can switch crop modules to build a DNA fingerprint database for a crop of interest. PIDS covers both individual detection and group detection, so it can be extended to DNA detection and genotyping applications in animals and microorganisms. This system has great universal adaptability and stable application effects.

## 3. Results

### 3.1. System Modules and Functionality

The core functions of PIDS include data generation, data storage, data audit, and data analysis. By providing automatic data generation, storage, audit, and rapid comparison functions, it can replace the previous methods of manually entering data into the database and manually comparing and merging data. Only a small amount of data needs to be corrected manually, namely data that the algorithm cannot automatically determine, to achieve the target of rapid processing of DNA fingerprint data. The data generation function in PIDS is divided into two parts, experimental information processing and fingerprint data analysis processing. These two parts correspond to the phases before and after a complete experiment, namely the experimental phase and the data analysis phase. Thus, PIDS provides comprehensive data analysis auxiliary functions for the experimenter, simplifies the often difficult data analysis phase, improves the quality of data analysis, and provides the basis for the analysis of mass fingerprint data. The modular structure of PIDS is shown in Figure 2.

PIDS can guide and assist users in completing genotyping experiments. Users can enter a project name to begin. The plate table holds information about specific plates (each with a unique barcode), so they can be individually tracked and the results recorded. The information in the plate table includes monitoring and allocating DNA sample locations in plates, loci/loci groups, panel, alleles, genotypes, upper template, PCR plate, electrophoresis plate, and primer information. Users can also upload capillary electrophoresis (CE) output files. Three types of data files can be uploaded: project file, Excel file, and GeneMapper output file. Project files need to be generated using the SSR Analyser software (downloadable free from https://ssr.PIDS.online:8445/ssr). This software can match a corresponding CE image on each primer locus as fingerprint data to upload to the server, and PIDS will automatically bind and store the image and primer locus. SSR Analyser runs on a Windows system that is automatically connected to the web services when opened from PIDS. For Excel files, PIDS provides an export format template file that can be used to generate data in the specified format.

PIDS uses Spring Web Services to handle incoming data files. The standard WSDL file is used to describe the related information Service interfaces and parameters. It contains the invocation specifications for the interfaces of three services (storage of fingerprint data, storage of image and panel file download). The WSDL file can be accessed at http://ssr.pids.online:6060/SsrDatasService/geneUpload.wsdl. For users who have technical problems trying to call this interface, we have provided an e-mail address where we can be contacted for help. We will consider providing an example to help users make smooth calls to the interface.

The Fingerprint Comparison Algorithm is applied to complete the comparison between the source and target fingerprints, which can reveal differences, missing, or no differences between fingerprints. PIDS focuses on the core identification function, including authenticity identification, purity identification, and paternity testing. It also has a genetic analysis function that allows users to perform genetic clustering and heterosis group analyses of their uploaded data.

PIDS has an identity authentication function when uploading, which prevents illegal data storage. Uploaded data can be viewed, downloaded, and printed. PIDS also generates reports for varieties identification and genetic analysis results. Users can perform genetic cluster analysis on the uploaded data with 12 genetic algorithms. PIDS also can produce common cluster tree images. The PIDS website provides users with detailed and comprehensive user guides. The detailed description of the examples is described in the PIDS User Manual and can be downloaded from https://ssr.pids.online:8445/ssr/sys/cms/files/download/6ec27a20252811ea2a6c538bc13df1b6 and the SSR Analyser User Manual can be downloaded from https://ssr.pids.online:8445/ssr/sys/cms/files/download/1a1b5ed011bd11e4b2a205874e1ede97).

### 3.2. System Model 

PIDS was constructed using a relational database. The database is implemented based on the current mainstream open source software MySQL. Figure 3 shows the entity relationship model (ERD) and the class table model in PIDS. Using Chen’s ERD notation to represent the ERD [31], we first identified 10 entities and four relationships (Figure 3A). A table-like model is constructed based on the ERD (Figure 3B). The “PCR plate” and “CE plate” entities shown in Figure 3A are each split into two tables, “PCR” and “PCR_well”, and “CE” and “CE_well” as shown in Figure 3B. These tables are used to include additional information to describe the wells in the plate and to accurately locate them. All the entities are related by the source of the sample and associated with basic information such as primers, panels, and detection equipment to build a complete fingerprint data information system.

The whole DNA fingerprint database contains basic information, experimental information, and fingerprint data information. These data are referenced to each other by IDs or bar code numbers. To solve the problem that fingerprint data are compatible with different crop primers, PIDS stores fingerprint data and fingerprint image information in independent files. The fingerprint data file is associated with the storage path information of the fingerprint image, and then the fingerprint data file path information is stored in the basic information table of fingerprint data. When loading and updating fingerprint data and fingerprint images, only new information needs to be written into the fingerprint data file. This approach avoids the problem of slow operations such as queries that use a database to store a large amount of binary data. Further, the fingerprint data and fingerprint image information are stored with greater freedom, and the DNA fingerprint database can be backed up and restored more quickly.

### 3.3. System Access

The PIDS interface focuses on simplicity as the basic design element. Front-end Web pages are built based on the open source UI framework Bootstrap3 (https://www.bootcss.com/) and some excellent open source JavaScript plug-ins have been incorporated to display page data to achieve the best user experience. In the basic process of PIDS, each operation step generally corresponds to a data batch import function to facilitate the completion of various key data warehousing operations. Excel and CSV are two common data formats in the whole system, which ensures fast data processing speeds.

As shown in Figure 4A, users can first click on the “Template” button to download a data sample template, and then write their self-organized data into this file according to the template requirements. When complete, the file can be imported into the system using the “Upload” or “Save” button. By following the basic steps shown in Figure 4A and the sequence of importing experimental data described in the PIDS user manual, users can complete the design process, including experiment design, data analysis, fingerprint and fingerprint image storage, fingerprint data viewing, and various identification and data analysis functions.

PIDS uses a standardized experimental design process (Figure 4A), so that experimenters have a unified experimental design language to communicate and understand each other. Further, the basic idea of “design before experiment”, ensures that all the experiments are adequately prepared and fully designed, and experimenters only need to carry out the experiment according to the printed electroplating layout table (Figure 4B) that is printed after the design process is complete. Thus, the experimental design and experimental operation are separated from the personnel to achieve a smooth pipeline operation. In the design process, the automatic electrophoresis plate design algorithm in PIDS is used to avoid the error and low efficiency of the sample hole position during manual typesetting.

After the data are stored in the database by the automatic binding of the fingerprint data and image process, users can view the most comprehensive overview of the fingerprint data at any time. For example, users can view the fingerprint data and CE image contained in all experiments of sample-associated DNA (Figure 4C). This helps users to determine the true cause of the fingerprint difference between different experiments, and ensures that the final integrated fingerprint data are sufficiently reliable and representative.

After the fingerprint data are entered into the local fingerprint database, DNA fingerprint comparisons can be performed to complete the identification of the variety. After entering the relevant parameters, such as the fingerprint range and the number of different loci of the fingerprint comparison, fingerprint data comparisons can be performed, and users can view statistical information and detailed comparison data information, such as difference, no difference, and missing among the fingerprints (Figure 4D).

PIDS also can provide a genetic analysis function based on the LFD fingerprint or user-defined fingerprint data through the genetic analysis function (Figure 4E). This function supports a variety of genetic distance calculation algorithms and genetic analysis graphic types, which can complete common graphical displays of multiple types of genetic analysis results, and provide functions that include genetic path, gene frequency table, and download of genetic analysis maps to allow users to obtain and save their own analysis results.

### 3.4. Data Quality Control

Data quality control is an important application of PIDS and the following methods are provided to complete this function.

Standardized experimental design method: The PIDS experimental design function is used to pre-plan the details of the sample testing process. This function provides a unified experimental design scheme for users to improve the standardization of their experimental design and to provide important reference information for quality control.

Add standard sample as reference: Experimenters can set one or more standard samples of the crops as reference samples during the experiment. By comparing the fingerprints of the reference samples with the fingerprint of the existing standard sample in PIDS, the system can determine whether there is an overall error in the current detection experiment.

Three repeated experiments: Three repeated experiments are carried out on the experiment to obtain multiple sets of fingerprint data. The automated fingerprint data audit function of PIDS is used to audit the results of the repeated experiments. The results of the differential loci determination are obtained after the audit. The difference and number of missing loci obtained from the results are used to analyze the consistency of the fingerprint data. The results indicate the basic quality of the experiment.

Basis for accuracy determination: Differences among loci can be comprehensively determined by the peak shape of all pending sample fingerprints. Because the fingerprints contain all the effective original information, they can be used as the most effective and accurate basis for data analysis by providing effective support for the quality of the final determination.

Thus, this platform provides functional support for data quality control from many aspects and dimensions. Data noise information generated during an experiment can be solved by the automated fingerprint audit of three repeated experiments, and the repeatability and stability of each locus in an experiment also can be determined. After the data noise has been processed, the number of different loci will increase, thereby providing a simple and effective basis for accuracy determination.

We prepared three sets of repeated experimental data for a maize hybrid sample; two of the sets are consistent and the other set is noise data. We uploaded the three sets of data to the PIDS system and carried out an automated fingerprint data audit through the menu functions “Experiment” and "Experiment Audit". We found that when the number of experiments was an odd number, using < 40% of the noise data did not cause adverse effects and the number of effective data loci did not reduce, whereas using ≥40% of the noise data caused differences in loci and missing data. The proportion of noise data can be adjusted, but because PIDS uses statistical methods to obtain the most consistent genotype of each locus, to ensure the noise data does not adversely affect the results, 40% noise data is set as the maximum threshold boundary for determining effective data.

## 4. Discussion

PIDS can process large amounts of DNA fingerprint data and perform quality control, which is urgently needed to track samples and analyze data. The extension of back-end functions was developed to meet the requirements of different laboratories. Currently, PIDS can support a variety of crops, including maize, rice, soybean, cotton, and other major food crops. PIDS also can support human and microbial DNA fingerprint data management and analysis. The functions of PIDS can be expanded according to the target samples studied. PIDS is potentially an efficient DNA fingerprint management platform that can be used for crop breeding research, variety protection, and research in the fields of medicine, crime, and microbiology. Although the primers used for different species will be different, the requirements for a DNA fingerprint database are similar [21,22]. Species type fields can be added to the sample information table to distinguish samples. Because all the markers are species specific, to ensure that the naming of the marker sites of different species do not conflict, a species type field is added to the marker information table to distinguish the markers. The same fingerprint database management system can be used for cotton and other polyploid species [9]. If the adopted markers have homologs on different chromosomes, the filtering algorithm of the SSR Analyzer and the structure of the fingerprint database management system will need to be adjusted.

In future updates, PIDS will be modified to support a variety of markers, including SSRs, SNPs, and indels. A marker type field will need to be added to the marker information table to distinguish the different markers, and the similarities and differences of different marker structures will need to be considered when designing the database structure. For example, for diploid species, only 1–2 alleles can be collected during data collection, but the alleles can be named according to the characteristics of the different markers. SSR markers can be named according to the size of the allele fragment, indel markers can be named according to the deletions/insertions as A and B, and SNP markers can be named according to the single base at the mutation position. For indel and SNP markers, a polyacrylamide gel electrophoresis (PAGE) detection platform can be used to manually collect data, which can be imported into the database as an Excel file. Fluorescent scanning solutions, such as chip detection platforms, also can be used. SNP markers need special tail sequences when used on electrophoresis detection platforms, whereas indel markers are relatively easy to use. When indel and SNP markers are created, there are still requirements for parallel experiments and layer upon layer merging.

Compatible platforms for PIDS are mainly PAGE and fluorescent capillary electrophoresis platforms. The compatibility of data collection methods, such as PAGE, which uses Excel tables, and capillary electrophoresis, which involves online fingerprint analysis and uploading of data and fingerprint image, needs to be considered. Different capillary electrophoresis platforms can be customized for sample plate design, electrophoresis plate design, and FSA file (ABI fragment analysis data file) configuration. The next generation of molecular identity cards can be linked to the PIDS website to establish a quick response (QR) code for each variety on the website. By scanning the QR code, users can retrieve molecular data information and fingerprint information of the varieties.

## 5. Conclusions

PIDS is an indispensable tool in our laboratory. It assists in automating DNA fingerprint experiments and reduces human error. It can complete sample tracking and perform common genetic analysis, thereby improving work efficiency and quality. PIDS can support all diploid plants and can be extended to support polyploid species. We can provide users with free customization and extension of back-end functions to meet the requirements of their laboratories, such as those involved in human and microorganism research. PIDS can monitor the experimental process and ensure the standardization of DNA fingerprint data. It can be used to conduct inter-database conversations, and exchange fingerprint data between fingerprint databases, with complete fingerprint data processing services. PIDS includes location statistics, fingerprint merging, fingerprint comparison, and genetic analysis functions, and is compatible with single and mixed DNA sample processing methods. PIDS has a complete loci statistics function that can meet the needs of a laboratory’s internal fingerprint database construction. PIDS can also meet the requirements for standard fingerprint database construction and sharing, and supports the expansion of multiple detection technologies and multiple fingerprint data services.

## Figures and Tables

**Figure 1 genes-11-00373-f001:**
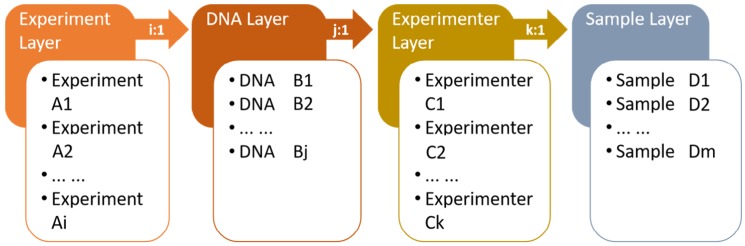
Schematic representation of the Fingerprint Merging Algorithm.

**Figure 2 genes-11-00373-f002:**
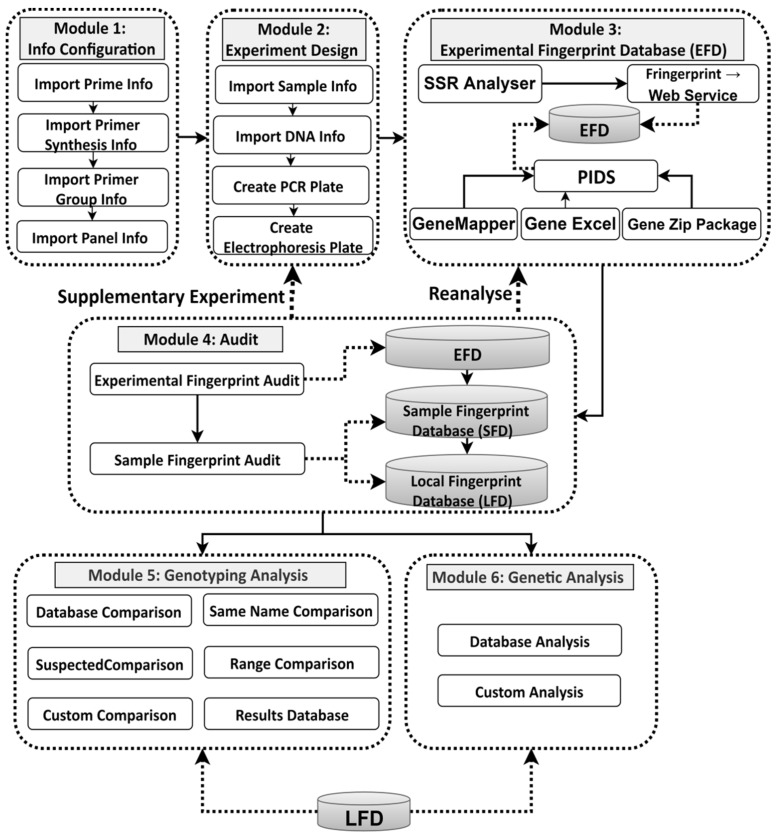
Modular structure of plant international DNA-fingerprinting system (PIDS), including the workflows between the modules and functions. PIDS covers three core fingerprint databases and the data transfer relationship between them. Modules 5 and 6 perform the core functions of identification and data analysis. The workflows ensure that the entire experiment and data analysis is a closed loop process. When problems occur, they can be solved by supplementary experiments or reanalysis, which can improve the efficiency of the data transfer process.

**Figure 3 genes-11-00373-f003:**
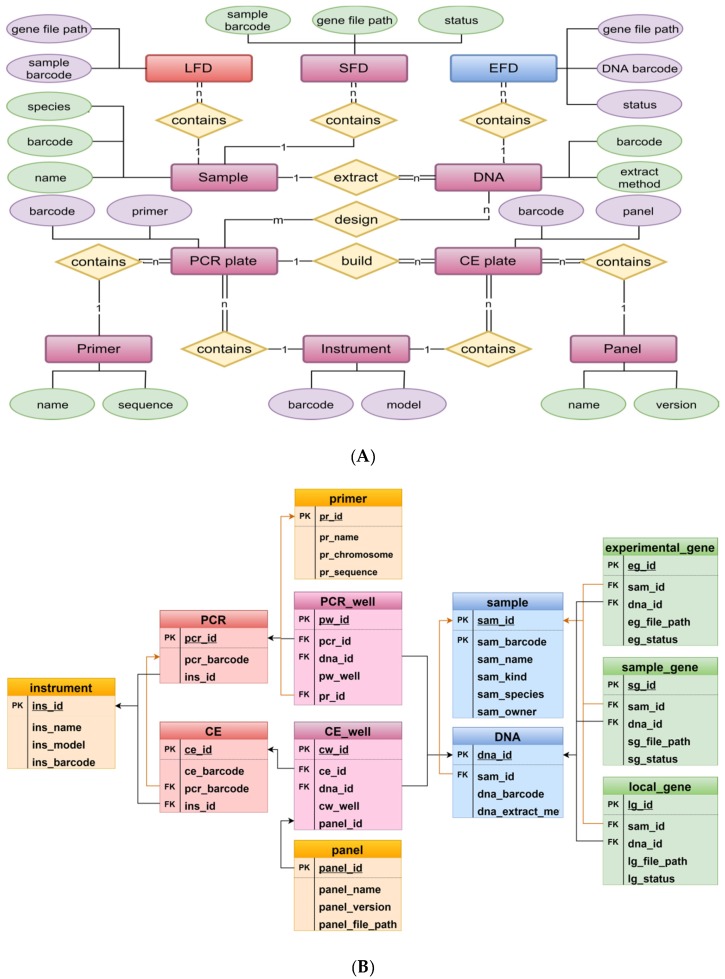
Entity relationship and class table models in PIDS. (**A**) Entity relationship model diagram (ERD) based on Chen’s ERD notation. The rectangular boxes indicate entities; ellipses indicate properties of the entities; and diamonds indicate relationships between entities. LFD, Local Fingerprint Database; SFD, Sample Fingerprint Database; EFD, Experimental Fingerprint Database. (**B**) Table-like model diagram based on the ERD. The top of each box contains the table name. Primary key (PK) indicates a primary key. Foreign key (FK) indicates a foreign key to another table and is indicated by an arrow.

**Figure 4 genes-11-00373-f004:**
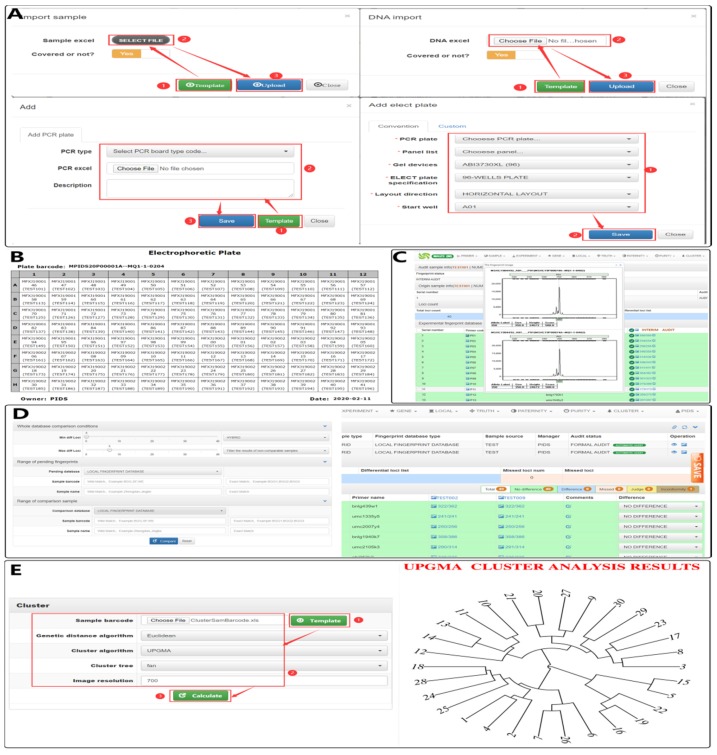
The PIDS interface and representative workflows. (**A**) Users can import the sample information table and DNA information table of the experiment, and automatically create PCR and electrophoresis plates according to the given condition parameters. (**B**) Users can print the automatically generated electrophoresis plate hole position information table for actual DNA detection experiments. (**C**) Users can view the fingerprint data and map information submitted for storage. (**D**) Users can enter comparison condition parameters and view detailed information of the comparison results. (**E**) Users can enter genetic analysis parameters and generate a chart of genetic analysis related results.

**Table 1 genes-11-00373-t001:** Conditions to be specified for the Fingerprint Comparison Algorithm.

Comparison Method	Default Fingerprint Data Range
Database Comparison	Entire Local Fingerprint Database
Homonymy Comparison	Fingerprints of the same name or synonyms within the entire Local Fingerprint Database
Non-homonymy Comparison	Fingerprints with different names and synonyms within the entire Local Fingerprint Database
Sub-Database Comparison	Assigned through Excel
Paired Comparison	Assigned through Excel

**Table 2 genes-11-00373-t002:** Parameters required in the Fingerprint Comparison Algorithm.

Condition	Description
Number of comparison loci	To control the matching of locus between different fingerprints, value: X ≥ 0. This parameter is used in the Fingerprint Comparison Algorithm, proper control of a value can reduce invalid comparison to improve fingerprint comparison speed, the default value of min(X) is 20.
Number of differential loci	To control the locus difference between samples, which used in the Fingerprint Comparison Algorithm to filter the comparison results. Proper control of a value can reduce the display of useless results, the range of X is ≥0, and the default value of max(X) is 20.
Percentage of differential loci	To control the degree of difference between samples, the range of X is 0 ≤ X ≤ 1, detail descriptions can be found in the mixed strain comparison algorithm. The default value of max(X) is 0.05.
Base offset	To control the difference between the two loci, the range of X is 0 bp ≤ X ≤ 2 bp. The default MaxX is 2 bp. This parameter is used in the comparison algorithm.

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
