# Peer review of "PIDS: A User-Friendly Plant DNA Fingerprint Database Management System"

_genes, 2020, doi:10.3390/genes11040373_

Round 1
Reviewer 1 Report
In this article, the authors describe an online platform they have developed to automate DNA fingerprinting in plants. Their platform, named Plant International DNA System (PIDS), provides an integrated framework to collect, analyze, and manage experimental data.
The article is reasonably clearly written, and its content is of interest to the readership of the journal. As such, I would recommend it to be accepted assuming the following concerns have been addressed.
Main concern: One of the main points of the platform is that it can deal with a large volume of data and perform quality control. However, no results are given for either of those claims. How large a volume are we considering here? Gigabytes? Terabytes? For quality control, the authors could artificially add noise to their data and see how robust their approach is.
Minor comments:
- The name of the platform (Plant International DNA System) is a bit bland. In particular, it should probably reference fingerprinting.
- Section 2.2: Algorithms are described textually only. However, it is standard to include pseudocode for algorithms.
- While Section 3 mentions that examples are available online, it would be helpful to the reader to have some added to the main text. The authors could expand on the global workflow shown in Figure 4.
- The SSR analyzer software used to prepare data only runs on Windows, reducing the potential reach of their platform. It would be helpful to potential users if the authors either provided source code, an online interface for the tool, or a detailed file format allowing the development of alternatives.
- In general, I feel that providing open-source code would be helpful to the community. While the authors promise that their platform can be enhanced by user extensions, nothing beats the possibility to interact directly with the code.
- Line 276, the authors mention "Chen's ERD notation". While that notation is fairly standard, they should still include the relevant reference: Chen, P. The Entity-Relationship Model - Toward a Unified View of Data. ACM Transactions on Database Systems. 1 (1): 9–36.
- The article contains numerous (small) English issues, such as mixing up "the" and "a", using incorrect word orders, or using informal language. While the text is written in a clear fashion, a professional editing service could add some polish to the text.
Author Response
Our point-by-point responses to reviewer #1’s comments
Those comments are all valuable and very helpful for revising and improving our paper, as well as the important guiding significance to our researches. Changes made to the text in response to the reviewer’s comments are highlighted in the revised manuscript.
Point 1: One of the main points of the platform is that it can deal with a large volume of data and perform quality control. However, no results are given for either of those claims. How large a volume are we considering here? Gigabytes? Terabytes?
Response 1: The data processing capacity of the International DNA-Fingerprinting System (PIDS) depends on the disk storage and memory capacities of the server hardware being used. Disk storage capacity determines how much detailed fingerprint data can be stored. Memory capacity determines how much fingerprint data can be loaded for comparison tasks. PIDS has been used widely by 20 institutions for more than five years, including the China National Rice Research Institute, the Chinese Academy of Agricultural Sciences , and the Development Center for Science and Technology, and has proved to be a stable efficient system for plant DNA detection and genetic analyses. At present, the largest number of fingerprint data sets stored by a single institution is over 700,000, which uses about 100 GB of storage capacity, and this is increasing year by year. PIDS can be implemented by extending computing performance by expanding memory and disk capacity and having a good multi-core CPU, which is usually sufficient for most laboratory needs.
Point 2: For quality control, the authors could artificially add noise to their data and see how robust their approach is.
Response 2: It is really true as Reviewer suggested. We agree. We have studied comments carefully and have made correction which we hope meet with approval. We added noise data to repeated experimental data for a maize hybrid sample and included the results in the Results section (lines 365–401).
Point 3: The name of the platform (Plant International DNA System) is a bit bland. In particular, it should probably reference fingerprinting.
Response 3: We agree. We renamed the system as the Plant International DNA-Fingerprinting System (PIDS) in the revised manuscript.
Point 4: Section 2.2: Algorithms are described textually only. However, it is standard to include pseudocode for algorithms.
Response 4: We are very sorry for our negligence of the algorithm description. We have done as suggested and provided the pseudocodes for the merging and comparison algorithms (lines 223).
Point 5: While Section 3 mentions that examples are available online, it would be helpful to the reader to have some added to the main text. The authors could expand on the global workflow shown in Figure 4.
Response 5: We have not included detailed examples in the main text because of the length of the manuscript. However, examples are described in detail in the PIDS User Manual.
Similarly, we have not expanded the global workflow shown in Figure 4. Because of the length of the manuscript, we illustrated only the core workflow. The complete global workflow and detailed description are given in the PIDS User Manual.
We have added links to the PIDS User Manual in the revised manuscript to help interested readers quickly access examples and workflow information (lines 276-280).
Point 6: The SSR analyzer software used to prepare data only runs on Windows, reducing the potential reach of their platform. It would be helpful to potential users if the authors either provided source code, an online interface for the tool, or a detailed file format allowing the development of alternatives. In general, I feel that providing open-source code would be helpful to the community. While the authors promise that their platform can be enhanced by user extensions, nothing beats the possibility to interact directly with the code.
Response 6: We are very sorry for our negligence of the description of SSR analyzer software. We have rewritten the description of SSR analyzer software in the revised manuscript (lines 260-266) and added links to the SSR Analyser User Manual (lines 278-280). PIDS uses Spring Web Services to handle incoming data files. The standard WSDL file is used to describe the related information Service interfaces and parameters. It contains the invocation specifications for the interfaces of three services (storage of fingerprint data, storage of image and panel file download). The WSDL file can be accessed at http://ssr.pids.online:6060/SsrDatasService/geneUpload.wsdl. For users who have technical problems trying to call this interface, we have provided an e-mail address where we can be contacted for help. We will consider providing an example to help users make smooth calls to the interface.
The relevant source code of SSR Analyser has been running for more than 5 years. It has analyzed more than 700,000 copies of data in the field of plant DNA fingerprint detection, and has applied it to more than 12 detection agencies and companies in China. The automated analysis algorithms for plants have also been professionally discussed and tested. This source code has been tested in a long-term actual production environment. The software itself is very stable and reliable, and it may not have room for subsequent optimization of algorithms and functions. Therefore, as a stable tool, readers are not advised to conduct modified on the source code. Thus, we will not release the corresponding source code for the time being. If readers are interested, they can request this part of the source code by email.
Point 7: Line 276, the authors mention "Chen's ERD notation". While that notation is fairly standard, they should still include the relevant reference: Chen, P. The Entity-Relationship Model - Toward a Unified View of Data. ACM Transactions on Database Systems. 1 (1): 9–36.
Response 7:. Special thanks to you for your good comments. We have added the reference as suggested (line 291 and 559).
Point 8: The article contains numerous (small) English issues, such as mixing up "the" and "a", using incorrect word orders, or using informal language. While the text is written in a clear fashion, a professional editing service could add some polish to the text.
Response 8: Thank you very much for your comments and suggestions. We checked the text carefully and also had it professionally edited.
Reviewer 2 Report
The manuscript is well written. The introduction provide sufficient background and include all relevant references, the methods are adequately described and the results Are clearly presented. the conclusions Are supported by the results.
Author Response
Our point-by-point responses to reviewer #2’s comments
Dear reviewer :
Thank you for your comments concerning our manuscript entitled “Plant DNA Fingerprint Database Management Using Plant International DNA System (PIDS)” (Manuscript ID: genes-737052). Those comments are all valuable and very helpful for revising and improving our paper, as well as the important guiding significance to our researches. We appreciate for your warm work earnestly.
Once again, thank you very much for your comments and suggestions.
Round 2
Reviewer 1 Report
The authors have addressed my comments in a satisfactory fashion, and I believe that the paper can be accepted in its current form.
Two minor comments:
- The pseudocode for the algorithms is appreciated. However, it lacks formatting. I had a quick look at the instructions for authors, but could not find specific formatting. Hopefully, the editor can provide further instructions. Otherwise, the authors could take inspiration from the guidelines of other communities, e.g., those of the ACM: page 5 of https://www.acm.org/binaries/content/assets/publications/taps/acm_layout_submission_template.pdf
- The quality of the English language in the text did improve overall, except in parts that were added for this round of review. I assume those parts were added or modified after proofreading. Again, that point is very minor, and the overall readability of the text isn't affected.
Author Response
Point 1: The pseudocode for the algorithms is appreciated. However, it lacks
formatting. I had a quick look at the instructions for authors, but
could not find specific formatting. Hopefully, the editor can provide
further instructions. Otherwise, the authors could take inspiration from
the guidelines of other communities, e.g., those of the ACM: page 5 of
https://www.acm.org/binaries/content/assets/publications/taps/acm_layout_submission_template.pdf.
Response 1: Those comments are all valuable and very helpful for revising and improving our paper. Changes made to the text in response to the reviewer’s comments are highlighted in the revised manuscript
(Line 223).
Point 2: The quality of the English language in the text did improve overall,
except in parts that were added for this round of review. I assume those
parts were added or modified after proofreading. Again, that point is
very minor, and the overall readability of the text isn't affected.
Response 2: Thank you very much for your comments and suggestions. We checked the text carefully and also had it professionally edited.